# A Probabilistic Matrix Factorization Method for Identifying lncRNA-Disease Associations

**DOI:** 10.3390/genes10020126

**Published:** 2019-02-08

**Authors:** Zhanwei Xuan, Jiechen Li, Jingwen Yu, Xiang Feng, Bihai Zhao, Lei Wang

**Affiliations:** 1College of Computer Engineering & Applied Mathematics, Changsha University, Changsha 410001, China; Zhanwei_xuan@163.com (Z.X.); lijiechen39555@163.com (J.L.); jingwen.yu18@gmail.com (J.Y.); fengxiang@xtu.edu.cn (X.F.); bihaizhao@163.com (B.Z.); 2Key Laboratory of Hunan Province for Internet of Things and Information Security, Xiangtan University, Xiangtan 411105, China

**Keywords:** lncRNA, disease, miRNA, lncRNA-disease associations, identifying disease-related lncRNA

## Abstract

Recently, an increasing number of studies have indicated that long-non-coding RNAs (lncRNAs) can participate in various crucial biological processes and can also be used as the most promising biomarkers for the treatment of certain diseases such as coronary artery disease and various cancers. Due to costs and time complexity, the number of possible disease-related lncRNAs that can be verified by traditional biological experiments is very limited. Therefore, in recent years, it has been very popular to use computational models to predict potential disease-lncRNA associations. In this study, we constructed three kinds of association networks, namely the lncRNA-miRNA association network, the miRNA-disease association network, and the lncRNA-disease correlation network firstly. Then, through integrating these three newly constructed association networks, we constructed an lncRNA-disease weighted association network, which would be further updated by adopting the KNN algorithm based on the semantic similarity of diseases and the similarity of lncRNA functions. Thereafter, according to the updated lncRNA-disease weighted association network, a novel computational model called PMFILDA was proposed to infer potential lncRNA-disease associations based on the probability matrix decomposition. Finally, to evaluate the superiority of the new prediction model PMFILDA, we performed Leave One Out Cross-Validation (LOOCV) based on strongly validated data filtered from MNDR and the simulation results indicated that the performance of PMFILDA was better than some state-of-the-art methods. Moreover, case studies of breast cancer, lung cancer, and colorectal cancer were implemented to further estimate the performance of PMFILDA, and simulation results illustrated that PMFILDA could achieve satisfying prediction performance as well.

## 1. Introduction

Long non-coding RNAs (lncRNAs) are a class of important heterologous ncRNAs that differ in length from miRNAs by more than 200 nucleotides [1]. For a long time, lncRNAs have been considered to be transcriptional noise, and only recently have these views been changed by increasing evidence [2]. Related studies have shown that lncRNA plays an indispensable role in many biological processes, such as chromatin remodeling, gene transcription, protein transport and trafficking, and epigenetic regulation [3,4,5,6,7,8,9]. In addition, the dysregulation of lncRNA in coronary artery disease, autoimmune disease, neurological disorder, and various cancers suggests that lncRNA plays an important role in many complex diseases [10]. Recently, lncRNAs are increasingly attracting the attention of researchers in the field of bioinformatics [10,11,12,13].

With the rapid development of high-throughput sequencing technology, thousands of lncRNAs have been discovered in mammalian transcriptions. Numerous studies have also revealed the important role of lncRNA in biological processes and the significant effects in complex human diseases [1]. There is no doubt that lncRNAs are closely related to complex human diseases, and more importantly, some lncRNA-disease associations have been experimentally confirmed. For example, the expression of XIST is up-regulated in glioma tissues and GSCs. Functionally, XIST knockdown exerts tumor suppressor function by reducing cell proliferation, migration and invasion, and inducing apoptosis [14]. LncRNA HOTAIR is highly expressed in prostate cancer and is associated with the growth and aggressiveness of prostate cancer cells [15]. Hence, it is meaningful to identify as many potential lncRNA-disease associations as possible. However, up to now, due to the high costs of traditional biological experiments, the lncRNA-disease associations supported by biological experiments are still very limited. Therefore, it is highly desirable to develop effective computational models to predict potential lncRNA-disease associations. In recent years, some computational models have been developed already, and all these models can be approximately divided into three different categories such as the machine learning-based models, biological network-based models and the models without relying on known lncRNA-disease associations [16].

As for the machine learning-based models, Chen Xing et al. proposed a computational model called LRSLDA to predict potential lncRNA-diseases associations [17] through acquiring two different scores from lncRNA space and disease space simultaneously for the same lncRNA-disease pair. Huang et al. proposed a prediction model called ILNCSIM by combining the LRSLDA, lncRNA functional similarity and disease semantic similarity to calculate the probabilities of lncRNA-disease associations [18]. Zhao et al. developed a Bayesian classifier-based model to identify new cancer-associated lncRNAs by using known cancer-associated lncRNAs such as multivariate data, genome, regulatory protein and transcription data integration [19].

As for the biological network-based models, based on the assumption that lncRNAs with similar functions are often associated with phenotype-like diseases, Sun et al. proposed a model called RWRlncD based on the lncRNA-lncRNA function similarity network [20]. Through integrating known lncRNA expression profiles, lncRNA-disease associations, lncRNA functional similarity, disease semantic similarity, and Gaussian interaction profile kernel similarity, Chen et al. developed a prediction model called KATZLDA to discover potential lncRNA-disease associations [21].

Among these machine-learning-based models and biological network-based models mentioned above, one of their common features is that known lncRNA-diseases relationships are required during the implement of prediction. However, so far, due to the time complexity and high costs of traditional biological experiments, the experimentally identified known lncRNA-disease associations are still very limited. Hence, some computational models that do not rely on known lncRNA-disease associations have been proposed in recent years. For instance, Liu et al. proposed a model based on the intermediate node genes to predict the potential disease-related lncRNAs [22]. Chen et al. proposed a model called HGLDA based on integrating miRNA-disease associations and lncRNA-miRNA interactions to discover novel lncRNA-disease associations [23].

In this paper, unlike the most advanced prediction models described above, a new model based on probability matrix decomposition called PMFILDA is proposed to discover potential lncRNA-disease associations. At present, matrix decomposition has been widely used in the field of bioinformatics. For example, in the prediction of miRAN-disease correlation, Chen et al. proposed to predict miRNA-disease correlation based on induction matrix complementation, matrix decomposition and heterogeneous graphs [24,25]. Zhao et al. proposed a method based on symmetric non-negative matrix factorization and Kronecker regularized least squares to predict the correlation of miRNA-disease [26]. The difference between our PMFILDA method and above-mentioned models is that we first constructed three kinds of binary association networks based on experimentally validated lncRNA-miRNA associations, miRNA-disease associations, and lncRNA-disease associations separately. Then, based on these three newly constructed association networks, we constructed a weighted lncRNA-disease association network. Moreover, based on the semantic similarity of disease and the functional similarity of lncRNA, we further adopted the KNN algorithm [27] to update the weighted lncRNA-disease association network. Then, according to the updated weighted lncRNA-disease association network, we decomposed the weight matrix of lncRNA-disease into low-order characteristic matrices U and V of the lncRNAs and diseases based on the probability matrix factorization. Finally, the product of U and V would be used to predict the scores of lncRNA-disease pairs. The flowchart of our prediction model PMFIDLA is shown in the following Figure 1.

In Subgraph A of above Figure 1, the lncRNA-miRNA association network is constructed based on known lncRNA-miRNA associations downloaded from starbase [28]. Nodes that are linked by solid lines indicate that they are associated. In Subgraph B, the miRNA-disease association network is constructed based on known miRNA-disease associations downloaded from HMDD [29]. Nodes that are linked by solid lines indicate that they are associated. In Subgraph C, the lncRNA-disease association network is constructed based on known lncRNA-disease associations downloaded from MNDR v2.0 [30]. Nodes that are linked by solid lines indicate that they are associated. In Subgraph D, the lncRNA-disease weight network is constructed based on Subgraph A, Subgraph B, and Subgraph C. Nodes that are linked by solid lines indicate that they are related. The blue dashed lines indicate the weight of the initial assignment between nodes. Subgraph E is a network of disease semantic similarity and the numbers in E are similarity scores. Subgraph F is a network of lncRAN functional similarity and the numerics in F are similarity scores. Subgraph G is a lncRNA-disease weighting network that has been updated and the red dashed line indicates weights having been redistributed between nodes. Subgraph H is the lncRNA-disease associations that are ultimately predicted by our method, and the solid red lines indicate the predicted lncRNA-disease associations with relatively high rankings. The KNN is a K-nearest neighbor algorithm used to find the most similar nodes. The PMF is a probability matrix factorization algorithm.

## 2. Materials

Since known lncRNA-disease associations were considered in our prediction model PMFIDLA, in this section, we download three kinds of gold standard datasets consisting of known lncRNA-miRNA associations, miRNA-disease associations, and lncRNA-disease associations from relevant authoritative databases, respectively.

### 2.1. Human LncRNA-MiRNA Associations and MiRNA-Disease Associations

Firstly, we downloaded the datasets of experimentally validated known miRNA-disease associations and lncRNA-miRNA associations from the two authoritative databases such as HMDD [29] and starbase [28] separately. Then, after having further unified the names of miRNAs in these two datasets, we could obtain 246 common miRNAs from both of these two datasets. For convenience, we denoted the set of these shared miRNAs as con_*M*. Thereafter, based on these 246 shared miRNAs in con_*M*, we finally downloaded 4704 different miRNA-disease associations and 9086 different lncRNA-miRNAs associations from above two authoritative databases. In addition, for convenience, we denoted the set of these 4704 different miRNA-disease associations as *MD* and the set of these 9086 different lncRNA-miRNAs associations as *LM* separately. Moreover, through statistics, there are 373 different diseases in *MD* and 1089 different lncRNAs in *LM* respectively (see Appendix A).

### 2.2. Human LncRNA-Disease Associations

Secondly, we downloaded the dataset of experimentally validated known lncRNA-disease associations from the MNDR v2.0 database [30], and for convenience, we denoted the dataset of these downloaded known lncRNA-disease associations as *LD*. Furthermore, to adapt the downloaded data to our prediction model, we would further process the original data as follows:**Step 1**:Obtaining the set of different lncRNAs shared in both *LD* and *LM*. In addition, for convenience, we denoted the set of these shared lncRNAs as con_*L*.**Step 2**:Obtaining the set of different diseases existed in *MD* and *LD*. In addition, for convenience, we denoted the set of these shared diseases as con_*D*.**Step 3**:Obtaining the set of lncRNA-disease associations with both lncRNAs in con_*L* and diseases in con_*D* based on the set of *LD*.

And as a result, we finally obtained 407 different lncRNA-disease associations including 77 different lncRNAs and 95 different diseases(see Appendix A).

## 3. Methods

### 3.1. Construction of the lncRNA-miRNA Association Network and miRNA-Disease Association Network

Based on the set of *LM* and *MD*, we can construct the lncRNA-miRNA association network and miRNA-disease association network according to the following steps respectively:**Step 1**:Supposing that there are $n_{l}$ different lncRNAs in *LM*, and for convenience, we denote the set of these lncRNAs as $L=\left\{l_{1},l_{2},\ldots ,l_{n_{l}}\right\}$.**Step 2**:Supposing that there are $n_{d}$ different diseases in *MD*, and for convenience, we denote the set of these diseases as $D=\left\{d_{1},d_{2},\ldots ,d_{n_{d}}\right\}$.**Step 3**:Supposing that there are $n_{m}$ different common miRNAs existed in both *MD* and *LM*, and for convenience, we denote the set of these miRNAs as $con\_M=\left\{m_{1},m_{2},\ldots ,m_{n_{m}}\right\}$.**Step 4**:Hence, we can firstly obtain an lncRNA-miRNA association network $G_{LMN}=\left(L,con\_M,E_{lm}\right)$, where $E_{lm}$ denotes the set of experimentally verified known associations in *LM*. For $\forall l_{i}\in L,m_{j}\in M$, we define that there is an edge $e_{l_{i}-m_{j}}$ between $l_{i}$ and $m_{j}$ in $E_{lm}$ if and only if there is an experimentally verified known associations between $l_{i}$ and $m_{j}$ in *LM*.**Step 5**:Simultaneously, we can also obtain an miRNA-disease association network $G_{MDN}=\left(con\_M,D,E_{md}\right)$, where $E_{md}$ denotes the set of experimentally verified known associations in *MD*. For $\forall m_{i}\in con\_M,d_{j}\in D$, we define that there is an edge $e_{m_{i}-d_{j}}$ between $m_{i}$ and $d_{j}$ in $E_{md}$ if and only if there is an experimentally verified known associations between $m_{i}$ and $d_{j}$ in *MD*.

### 3.2. Construction of the Weighted lncRNA-Disease Association Network

Based on the newly constructed association networks such as $G_{LMN}$ and $G_{MDN}$, we can further obtain a weighted lncRNA-disease association network $G_{LDWN}=\left(L,D,E_{ld},W_{ld}\right)$, where $E_{ld}$ denotes the set of edges between different lncRNAs in *L* and diseases in *D*. For $\forall l_{i}\in L,d_{j}\in D$, we define that there is an edge $e_{l_{i}-d_{j}}$ between $l_{i}$ and $d_{j}$ in $E_{ld}$ if and only if there is at least one miRNA $m_{k}$ in con_*M* with experimentally verified known associations with both $l_{i}$ in *LM* and $d_{j}$ in *MD* simultaneously. In addition, $W_{ld}=\left\{w_{l_{i}-d_{j}}|l_{i}\in L,d_{j}\in D,e_{l_{i}-d_{j}}\in E_{ld}\right\}$ denotes the set of weight of the edge $e_{l_{i}-d_{j}}$ in $E_{ld}$, and for $\forall l_{i}\in L,d_{j}\in D$, if there is $e_{l_{i}-d_{j}}\in E_{ld}$, then the weight $w_{l_{i}-d_{j}}$ corresponding to $e_{l_{i}-d_{j}}$ can be calculated according to the following steps:**Step 1**:Supposing that there are *T* different miRNAs in con_M with experimentally verified known associations with both $l_{i}$ in *LM* and $d_{j}$ in *MD* simultaneously, and for convenience, we denote the set of these *T* different miRNAs as $CM=\{m_{1},m_{2},\ldots ,m_{T}\}$.**Step 2**:Supposing that $RM_{l_{i}}=\left\{m_{l_{i1}},m_{l_{i2}},\ldots ,m_{l_{ip}}\right\}$ is a set consisting of all miRNAs that have experimentally verified known associations with $l_{i}$ in *LM*, and $RM_{d_{j}}=\left\{m_{d_{j1}},m_{d_{j2}},\ldots ,m_{d_{jq}}\right\}$ is a set consisting of all miRNAs that have experimentally verified known associations with $d_{j}$ in *MD*.**Step 3**:Let $RM=RM_{l_{i}⋃RM_{d_{j}}}=\left\{m_{1},m_{2},\ldots ,m_{S}\right\}$, then we can calculate the weight of $e_{l_{i}-d_{j}}$ in $G_{LDWN}$ according to the following Formula (1):
(1)$$w_{l_{i}-d_{j}}=\left\{\begin{array}{cc} 1 & \mathrm{if}l_{i}\mathrm{associated}\mathrm{with}d_{j}\mathrm{in}LD\\ T/(S+1) & \mathrm{Otherwise} \end{array}\right.$$

### 3.3. Similarity Calculation

#### 3.3.1. Disease Semantic Similarity Measure

Considering that the similarity between disease pairs can calculated by their directed acyclic graphs (DAGs) [31], while estimating the semantic similarity of diseases, for any given disease, we will firstly express it as its directed acyclic graph (DAG), and as illustrated in the following Figure 2, in its corresponding DAG, all annotated terms associated with this disease will be contained. For instance, in Figure 2 the DAGs of two different diseases such as Breast Neoplasms ($d_{1}$) and Liver Neoplasms ($d_{2}$) are shown, and it is obvious that the DAG of $d_{1}$ can be denoted as $DAG_{d_{1}}=\left(d_{1},T_{d_{1}},E_{d_{1}}\right)$, where $T_{d_{1}}$ denotes all the ancestor nodes of “$d_{1}$” and itself, and $E_{d_{1}}$ represents the set of edges in $DAG_{d_{1}}$. Moreover, for any disease $d^{\prime }$ in $DAG_{d_{1}}$, its semantic contribution to $d_{1}$ can be calculated according to the following Formula (2): (2)$$D_{d_{1}}\left(d^{\prime }\right)=\left\{\begin{array}{cc} 1 & \mathrm{if}\mathrm{there}\mathrm{is}d^{\prime }=d_{1}\mathrm{in}{\mathrm{DAG}}_{d_{1}}\\ max\{\Delta \times D_{d_{1}}\left(d^{\prime \prime }\right) & \mathrm{Otherwise}.\mathrm{Here}d^{\prime \prime }\in \mathrm{children}\mathrm{of}d^{\prime }\mathrm{in}{\mathrm{DAG}}_{d_{1}} \end{array}\right.$$
where Δ will be set to 0.5 based on the suggestion proposed by the state-of-the-art literature [31]. Moreover, in the same way, it is easy to see that the *DAG* of $d_{2}$ can be denoted as $DAG_{d_{2}}=\left(d_{3},T_{d_{3}},E_{d_{3}}\right)$, and then, for any given diseases $d_{1}$ and $d_{2}$, the semantic similarity between them can be measured according to the following Formula (3) obviously:(3)$$SD\left(d_{1},d_{2}\right)=\frac{\sum_{t\in T_{d_{1}}\cap T_{d_{2}}}\left(D_{d_{1}}\left(t\right)+D_{d_{2}}\left(t\right)\right)}{\sum_{t\in T_{d_{1}}}D_{d_{1}}\left(t\right)+\sum_{t\in T_{d_{2}}}D_{d_{2}}\left(t\right)}$$

#### 3.3.2. LncRNA Similarity Measure

The functional similarity between lncRNAs measures how similar their functions will be. In this section, based on the method proposed by the state-of-the-art literature [31], for any given lncRNAs $l_{i}$ and $l_{j}$, Supposing that $l_{i}$ and $l_{j}$ have known associations with a group of diseases $GD\left(l_{i}\right)=d_{i1},d_{i2},\ldots ,d_{ip}$ and $GD\left(l_{j}\right)=d_{j1},d_{j2},\ldots ,d_{jq}$ in *LD* respectively, then the functional similarity between them can be measured according to the following Formula (4):(4)$$FS\left(l_{i},l_{j}\right)=\frac{\sum_{t=1}^{p}S\left(d_{it},GD\left(l_{j}\right)\right)+\sum_{t=1}^{q}S\left(d_{jt},GD\left(l_{i}\right)\right)}{p+q}$$
(5)$$S\left(d_{k},GD\left(l_{i}\right)\right)=\max_{t\in [1,|GD\left(l_{i}\right)|]}SD\left(d_{k},d_{t}\right)$$

#### 3.3.3. Weight Redistribution in $G_{LDWN}$ Based on the KNN Algorithm

Based on the above descriptions, it is easy to see that we can represent the network $G_{LDWN}$ with its weight matrix $W_{ld}$, where $W_{ld}\left[i\right]\left[j\right]=w_{l_{i}-d_{j}}$. Moreover, considering that known lncRNA-disease associations are very sparse, which may cause that there exist some lncRNAs with no associations with any diseases, or some diseases with no associations with any lncRNAs. Hence, some potential associations between predicted lncRNAs and diseases will be invalid. Therefore, in this paper, we will rebuild the weight matrix $W_{ld}$ to solve this kind of problem as follows:**Step 1**:Firstly, representing the *ith* row of the weight matrix $W_{ld}$ as $W_{ld}\left(l_{i},:\right)=\left\{w_{l_{i}-d_{1}},w_{l_{i}-d_{2}},\ldots ,w_{l_{i}-d_{nd}}\right\}$, and the *jth* column of the weight matrix $W_{ld}$ as $W_{ld}\left(:,d_{j}\right)=\left\{w_{l_{1}-d_{j}},w_{l_{2}-d_{j}},\ldots ,w_{l_{nl}-d_{j}}\right\}$.**Step 2**:Then, for any given lncRNA $l_{q}$ and any $l_{i}$ in *L* other than $l_{q}$, based on above Formula (4), it is obvious that we can obtain the functional similarity $FS(l_{i},l_{q})$ easily, and moreover, after sorting these values of functional similarities between $l_{q}$ and all remaining lncRNAs other than $l_{q}$ in descending order, then we can obtain the corresponding lncRNAs from the first *K* elements in the sorted results. For convenience, let $l_{1},l_{2},\ldots ,l_{K}$ denote these *K* lncRNAs, then the *qth* row of $W_{ld}$ can be updated according to the following Formula (6):
(6)$$W_{ld}\left(l_{q},:\right)=\frac{1}{NL}\sum_{i\in [1,K]}\alpha^{i-1}*FS\left(l_{i},l_{q}\right)*W_{ld}\left(l_{i},:\right)$$
where α∈(0,1] is a decay factor, which means that a higher decay will be assigned to $l_{i}$ if it is more dissimilar to $l_{q}$, and $NL=\sum_{i\in [1,K]}FS\left(l_{i},l_{q}\right)$ is the normalization factor, which is used for normalization of the value of $W_{ld}\left(l_{q},:\right)$. Additionally, in similar way, it is obvious that the *pth* column of $W_{ld}$ can also be updated according to the following Formula (7):
(7)$$W_{ld}\left(:,d_{p}\right)=\frac{1}{ND}\sum_{i\in [1,K]}\beta^{i-1}*SD\left(d_{i},d_{p}\right)*W_{ld}\left(:,d_{p}\right)$$
where $d_{1}$ to $d_{K}$ denote the top *K* diseases most similar to $d_{p},\beta \in \left(0,1\right]$ is the decay factor, and $ND=\sum_{i\in [1,K]}SD\left(l_{i},l_{q}\right)$ is the normalization factor.

### 3.4. Construction of Our Prediction Model PMFILDA Based on $G_{LDWN}$

#### 3.4.1. Standard Matrix Factorization

Up to now, the matrix decomposition technology is widely used in the field of recommended systems, since not only the computational complexity can be reduced by matrix decomposition, but also good performance can be achieved in solving the matrix scarcity problem. The standard matrix decomposition aims to find two low-ranking, latent feature matrices whose products are used to fit the original matrix. Hence, for the weight matrix $W_{ld}\in R^{n_{l}\times n_{d}}$ constructed above, it is obvious that we can decompose $W_{ld}$ into two different matrices $U\in R^{n_{l}\times k}$ and $V\in R^{n_{d}\times k}$
$\left(k\ll min\left(n_{l},n_{d}\right)\right)$, and there is $W_{ld}\approx UV^{T}$. Thereafter, the problem of disease-related lncRNA prediction can be further expressed by the following Formulas (8) and (9):(8)$$\arg\min_{U,V}\sum_{i=1}^{n_{l}}\sum_{j=1}^{n_{d}}{\left(W_{ld}\left(i,j\right)-{\hat{W}}_{ld}\left(i,j\right)\right)}^{2}$$
(9)$${\hat{W}}_{ld}\left(i,j\right)=\sum_{k}U_{ik}*V_{jk}=\sum_{k}U_{i,k}*V_{kj}^{T}=U_{i}V_{j}^{T}$$
where the row vectors $U_{i}$ and is $V_{j}$ represent the ith lncRNA-specific and jth disease-specific latent feature vectors respectively. In addition, obviously, the above Formulas (8) and (9) form a convex optimization problem, which can be solved by some existing optimization algorithms such as the iterative update algorithm [32] easily.

#### 3.4.2. Probabilistic Matrix Factorization

Since the probability matrix decomposition is based on the decomposition of the standard matrix, supposing that $W_{ld}$ is a positive distribution with Gaussian noise, then we can define the conditional distribution over the $W_{l}d$ as: (10)$$p\left(W_{ld}|U,V,\sigma^{2}\right)=\prod_{i=1}^{n_{l}}\prod_{j=1}^{n_{d}}{\left[N\left(W_{ld}\left(i,j\right)|U_{i}V_{j}^{T},\sigma^{2}\right)\right]}^{I_{ij}}$$
where $N(W_{ld}\left(i,j\right)|U_{i}V_{j}^{T},\sigma^{2})$ is the probability distribution function of the normal distribution and $I_{ij}=\left\{\begin{array}{cc} 1 & W_{\mathrm{ld}}\left(i,j\right)\neq 0\\ 0 & \mathrm{Otherwise}. \end{array}\right.$ Obviously, $p(W_{ld}|U,V,\sigma^{2})$ is the likelihood function (i.e., the product of all the weights).

In addition, supposing that the matrices *U* and *V* satisfy the Gaussian prior to a mean of 0, then the priors of *U* and *V* can be denoted as follows:(11)$$p(U|\sigma_{U}^{2}=\Pi_{i=1}^{n_{l}}N\left(U_{i}|0,\sigma_{U}^{2}I\right))$$
(12)$$p(V|\sigma_{V}^{2}=\Pi_{i=1}^{n_{d}}N\left(V_{j}|0,\sigma_{V}^{2}I\right))$$

Here, the matrix *I* is a T×T dimensional unit diagonal matrix. Assuming that *U* and *V* are independent of each other, then the posterior distribution of *U* and *V* can be obtained by following Formula (13):(13)$$\begin{array}{c} p\left(U,V|W_{ld},\sigma^{2},\sigma_{U}^{2},\sigma_{V}^{2}\right)=\frac{p\left(W_{ld}|U,V,\sigma^{2},\sigma_{U}^{2},\sigma_{V}^{2}\right)\times p\left(U,V\right)}{p(W_{ld}|U,V,\sigma^{2},\sigma_{U}^{2},\sigma_{V}^{2})}\\ ∼p\left(W_{ld}|U,V,\sigma^{2},\sigma_{U}^{2},\sigma_{V}^{2}\right)\times p\left(U,V\right)=p\left(W_{ld}|U,V,\sigma^{2},\sigma_{U}^{2},\sigma_{V}^{2}\right)\times p\left(U\right)\times P\left(V\right)\\ =\Pi_{i=1}^{n_{l}}\Pi_{j=1}^{n_{d}}{\left[N\left(W_{ld}\left(i,j\right)|U_{i}V_{j}^{T},\sigma^{2}\right)\right]}^{I_{ij}}*\Pi_{i=1}^{n_{l}}N\left(U_{i}|0,\sigma_{U}^{2}I\right)*\Pi_{j=1}^{n_{d}}N\left(V_{j}|0,\sigma_{V}^{2}I\right) \end{array}$$

Then the log of the posterior distribution over the features of lncRNAs and diseases can be calculated as follows:(14)$$\begin{array}{c} \ln p\left(U,V|W_{ld},\sigma^{2},\sigma_{U}^{2},\sigma_{V}^{2}\right)=\sum_{i=1}^{n_{l}}\sum_{j=1}^{n_{d}}I_{ij}\ln N\left(W_{ld}\left(i,j\right)|U_{i}V_{j}^{T},\sigma^{2}\right)+\sum_{i=1}^{n_{l}}\ln N\left(U_{i}|0,\sigma_{U}^{2}I\right)\\ +\sum_{i=1}^{n_{d}}N\left(V_{j}|0,\sigma_{V}^{2}I\right)=-\frac{1}{2\sigma^{2}}\sum_{i=1}^{n_{l}}\sum_{j=1}^{n_{d}}I_{ij}{\left(W_{ld}\left(i,j\right)-U_{i}V_{j}^{T}\right)}^{2}-\frac{1}{2\sigma_{U}^{2}}\sum_{i=1}^{n_{l}}U_{i}U_{i}^{2}-\frac{1}{2\sigma_{V}^{2}}\sum_{j=1}^{n_{d}}V_{j}V_{j}^{2}\\ -\frac{1}{2}\left(\left(\sum_{i=1}^{n_{l}}\sum_{j=1}^{n_{d}}I_{ij}\right)\ln\sigma^{2}+Tn_{l}\ln\sigma_{U}^{2}+Tn_{d}\ln\sigma_{V}^{2}\right)+C \end{array}$$

Here, *C* is a constant factor. In addition, as for $N(U_{i}|0,\sigma_{U}^{2}I)$, since there is:(15)$$N\left(U_{i}|0,\sigma_{U}^{2}I\right)=-\frac{1}{{\left(2\pi \right)}^{\frac{T}{2}}{\left|\sigma_{U}^{2}I\right|}^{\frac{1}{2}}}exp\left(-\frac{1}{2}U_{i}{\left(\sigma_{U}^{2}I\right)}^{-1}U_{i}^{T}\right)$$

Hence, considering that the matrix ***I*** is an unit diagonal matrix, which means that there is ${\left(\sigma_{U}^{2}I\right)}^{-1}=\frac{1}{\sigma_{U}^{2}}I$, then we have:(16)$$lnN\left(U_{i}|0,\sigma_{U}^{2}I\right)=ln\left(-\frac{1}{{\left(2\pi \right)}^{\frac{T}{2}}{\left|\sigma_{U}^{2}I\right|}^{\frac{1}{2}}}\right)-\frac{U_{i}U_{i}^{T}}{2\sigma_{U}^{2}}=-\frac{T}{2}ln\left(\sigma_{U}^{2}\right)-\frac{U_{i}U_{i}^{T}}{2\sigma_{U}^{2}}+C_{U}$$

Here, $C_{U}$ is a constant factor. Similar to the above analyses, we can also have:(17)$$lnN\left(V_{j}|0,\sigma_{V}^{2}I\right)=-\frac{T}{2}ln\left(\sigma_{V}^{2}\right)-\frac{V_{i}V_{j}^{T}}{2\sigma_{V}^{2}}+C_{V}$$
(18)$$lnN\left(W_{ld}\left(i,j\right)|U_{i}V_{j}^{T},\sigma^{2}\right)=-\frac{1}{2}ln\left(\sigma^{2}\right)-\frac{{\left(W_{ld}\left(i,j\right)-U_{i}V_{j}^{T}\right)}^{2}}{2\sigma^{2}}+C_{W}$$

Therefore, it is obvious that maximizing the log-posterior on *U* and *V* with hyper-parameters being kept fixed in Formula (13) will be equivalent to minimizing the following objective function:(19)$$\arg\min_{U,V}\frac{1}{2}{\left||I⨀\left(W_{ld}-UV^{T}\right)|\right|}_{F}^{2}+\frac{\lambda_{U}}{2}\sum_{i=1}^{n_{l}}{\left||U_{i}|\right|}_{F}^{2}+\frac{\lambda_{V}}{2}\sum_{i=1}^{n_{d}}{\left||V_{j}|\right|}_{F}^{2}$$
where ⨀ is the Hadamard product, $\lambda_{U}=\frac{\sigma^{2}}{\sigma_{U}^{2}},\lambda_{V}=\frac{\sigma^{2}}{\sigma_{V}^{2}}$ and $\left|\right|\cdot {\left|\right|}_{F}$ represents the Frobenius norm.

#### 3.4.3. Optimization

Based on the properties of Frobenius norm, the Formula (19) can be rewritten as the form of the LaGrangian function as follows:(20)$$\begin{array}{c} L_{f}\left(U,V\right)=\frac{1}{2}Tr\left(I⨀\left(W_{ld}W_{ld}^{T}-2W_{ld}VU^{T}+UV^{T}VU^{T}\right)\right)\\ +\frac{\lambda_{U}}{2}Tr\left(UU^{T}\right)+\frac{\lambda_{V}}{2}Tr\left(VV^{T}\right) \end{array}$$

Based on above Formula (20), we can further obtain its partial derivatives with respect to *U* and *V* as follows:(21)$$\frac{\partial L_{f}}{\partial U}=I⨀\left(-W_{ld}V+UV^{T}V\right)+\lambda_{U}U$$
(22)$$\frac{\partial L_{f}}{\partial V}=I⨀\left(-W_{ld}^{T}U+VU^{T}U\right)+\lambda_{V}V$$

Therefore, we can construct the update rules based on the gradient descent algorithm as follows:(23)$$U\leftarrow \lambda_{m}U-\lambda \left(I⨀\left(-W_{ld}V+UV^{T}V\right)\right)$$
(24)$$V\leftarrow \lambda_{m}V-\lambda \left(I⨀\left(-W_{ld}^{T}U+VU^{T}U\right)\right)$$
where $\lambda_{m}$ is the momentum parameter, which can accelerate the convergence speed of *U* and *V*, the parameter λ denotes the learning rate, and based on the suggestion proposed by the state-of-the-art literature [31] (Wang et al., 2010), $\lambda_{m}$ and λ will be set to 0.8 and 0.005 respectively [33].

Hence, based on above update rules illustrated in Formulas (23) and (24), we can update the lncRNA-specific and disease-specific latent feature matrix *U* and *V* until they become converged. Then, we can finally obtain the predicted lncRNA-disease association matrix ${\hat{W}}_{ld}=FS\times UV^{T}\times SD$. In addition, as for any column $d_{i}$ in $W_{ld}$, we can sort the elements (i.e., lncRNAs) in $d_{i}$ in descending order, then the top-ranked lncRNAs in $d_{i}$ can be predicted as $d_{i}$-related lncRNAs, while the bottom-ranked lncRNAs in $d_{i}$ can be predicted as $d_{i}$- disrelated lncRNAs at the same time.

## 4. Results and Discussion

### 4.1. Performance Evaluation Metrics

To evaluate the robustness and prediction performance of PMFILDA, in this section, the Leave One Out Cross-Validation (LOOCV) was implemented based on the experimentally verified lncRNA-disease associations. In LOOCV, each pair of known lncRNA-disease associations is used as a validation set, while other known lncRNA-disease associations are used as training sets. Moreover, all the lncRNA-disease pairs without experimentally verify are used as candidate samples. The ranking of the test sample relative to the candidate sample needs to be evaluated after the implementation of PMFILDA. When a threshold is given, if the test sample ranks above the given threshold, then we will regard that a correctly positive sample has been predicted by PMFILDA, otherwise we will regard that a correctly negative sample has been predicted by PMFILDA. Moreover, while different thresholds are set, a series of True Positive Rate (TPR) and False Positive Rate (FPR) can also be obtained according to the following formulas:(25)$$TPR=\frac{TP}{TP+FN}$$
(26)$$FPR=\frac{FP}{TN+FP}$$
where *TP* and *TN* denote the number of positive and negative samples that have been correctly identified, while *FP* and *FN* represent the number of positive and negative samples that have been incorrectly identified. Hence, the Receiver Operating Characteristic (ROC) curve can be drawn by plotting TPRs versus FPRs, and the area under ROC curve (AUC) can be further calculated to measure the global performance of PMFILDA. Obviously, the closer the value of AUC is to 1, the more robust the prediction model would be.

Moreover, during simulation, to eliminate the random errors caused by the random initialization of *U* and *V*, we repeated our experiments 100 times and took the mean and variance of AUCs as our final results, which were shown in the following Figure 3. In addition, from Figure 3, it is easy to see that our newly proposed prediction model PMFILDA can achieve the mean AUC of 0.8794 and the standard deviation of 0.0011.

Next, to further evaluate the performance of PMFILDA, based on the framework of LOOCV, we compared PMFILDA with some state-of-the-art models such as NBCLDA [34], HGLDA [23], and the method proposed by Yang et al. [35]. Similarly, during simulation, to eliminate the random errors caused by the random initialization of *U* and *V*, we repeated our experiments 50 times and took the mean of AUCs as our final results, which were shown in the following Table 1.

In addition, while comparing PMFILDA with the NBCLDA, considering that we did not consider the genes in our method, then we compared PMFILDA with the $NBCLDA\_GN_{1}$ only, and the simulation results are shown in Table 1 and the following Figure 4. Obviously, from Table 1 and Figure 4, it is easy to see that our newly proposed prediction model PMFILDA can achieve a reliable AUC of 0.8793 that is much higher than the AUC of 0.8519 achieved by $NBCLDA\_GN_{1}$.

Moreover, while comparing PMFILDA with the HGLDA, in order to make a fair comparison, we implemented LOOCV on both PMFILDA and HGLDA based on the same dataset, i.e., we used the same 183 known lncRNA-disease associations proposed by HGLDA in the comparison simulation, and the simulation results are shown in Table 1 and the following Figure 5. Obviously, from Table 1 and Figure 5, it is easy to see that our newly proposed prediction model PMFILDA is superior to HGLDA.

Finally, while comparing PMFILDA with the method proposed by Yang et al, in order to make a fair comparison, we implemented LOOCV on both PMFILDA and method proposed by Yang et al. based on the same dataset also, i.e., we used the same 319 known lncRNA-disease associations between 37 lncRNAs and 52 diseases proposed by Yang et al in the comparison simulation, and the simulation results are shown in Table 1 and the following Figure 6. Obviously, from Table 1 and Figure 6, it is easy to see that our newly proposed prediction model PMFILDA can achieve a reliable AUC of 0.9090 that is much higher than the AUC of 0.8568 achieved by Yang et al.

### 4.2. Contribution Analysis of lncRNA-Disease Associated Network

In our method, we constructed a weighted lncRNA-disease association networks based on the known lncRNA-disease, microRNA-disease and lncRNA-microNA association networks. It may be useful to discuss the contribution of lncRNA-disease associations separately here. Hence, without considering the relationship between lncRNA and disease, we constructed a weighted network of lncRNA-disease through using known lncRNA-microRNA associations and microRNA-disease associations only. Based on the steps in Section 3.2, we finally obtained 304 lncRNA-disease associations including 60 lncRNAs and 73 diseases. Thereafter, we further obtained the corresponding weight matrix $W_{ld}$, and then performed LOOCV 100 times on the PMFILDA. The results were shown in the following Figure 7 and Figure 8, obviously, the AUC value achieved by PMFILDA based on three association networks can be increased by 0.0763 than the AUC value achieved by PMFILDA based on two association networks.

### 4.3. The Effects of KNN on Performance

Considering that known lncRNA-disease associations are very sparse, there may exist some lncRNAs with no associations with any diseases, or some diseases with no associations with any lncRNAs. Hence, some potential associations between predicted lncRNAs and diseases will be invalid. Therefore, in this paper, we rebuilt the weight matrix $W_{ld}$ based on KNN algorithm to solve this kind of problem.

Here, we also investigated the influence of KNN algorithm on our method from two aspects. One is that we do not use KNN algorithm to deal with the weighted network $G_{LDMN}$, the other is to update the weighted network $G_{LDMN}$ with other algorithms, such as K-means algorithm. When PMFILDA is directly executed without using KNN algorithm to process the weighted network $G_{LDMN}$, the result is shown in Figure 9. It is easy to see that PMFILDA could achieve an average AUC of 0.8794 while the weight of $G_{LDMN}$ was reallocated; however, while the weight of $G_{LDMN}$ was not reallocated, PMFILDA can achieve an average AUC of 0.8042 only, which demonstrated that it can improve the performance of our model through adopting the KNN algorithm to re-allocate the weight of $G_{LDMN}$.

And in addition, to estimate the impacts of other algorithms, we selected the K-means algorithm for further testing. After performing LOOCV 100 times, we presented the simulation results in the following Table 2, and from observing the results in Table 2, it is easy to see that the performance of KNN is better than K-means.

### 4.4. Parameter Sensitivity Analysis

From above descriptions, it is easy to see that to improve the prediction performance of PMFILDA, some parameters have been introduced in the model construction of PMFILDA, whose values will need to be finalized by the training of the prediction model. For example, how to choose the value of the parameter K while adopting the algorithm of K-nearest neighbor? How to choose the attenuation coefficients α and β given in the Formulas (6) and (7)? How to choose the value of the parameter T while adopting Formula (1) to implement the matrix decomposition? and so on. Hence, firstly, to evaluate the impacts of the parameters *K*, α and β to the performance of our model PMFILDA, during simulation, we will set K from 2 to 10 and α from 0.1 to 0.9, respectively. Moreover, we will set α=β for convenience. The detailed simulation results were shown in the Appendix A. In addition, through experimental results, it is easy to know that our model PMFILDA can achieve the highest AUC of 0.9204 while K=10 and α=β=0.8 in the LOOCV framework. Next, to estimate the impacts of the parameter T to the performance of our model PMFILDA, during simulation, we will set T from 10 to 50 and the step size to 10. In addition, through experimental results, it is easy to know that our model PMFILDA can achieve the highest AUC of 0.9210 while T=20 in the LOOCV framework.

### 4.5. Case Studies

In this section, we implemented case studies based on the optimal settings of above parameters to further verify the prediction performance of PMFILDA. During simulation, for each given disease, its potentially relevant lncRNAs predicted by PMFILDA will be sorted according to their predicted scores in descending order. In addition, as a result, the top 20 predicted lncRNAs related to the disease potentially will be recorded in the Appendix A, and then, two public databases such as MNDR V2.0 and LncRNA-Disease database will be used to confirm these potential associations between the given disease and each of these 20 predicted lncRNAs. In this section, we selected three kinds of common diseases such as breast cancer, lung cancer, and colorectal cancer as the targets of our case studies.

As for breast cancer, according to the reports of relevant literatures, it is very common in the group of women [36,37] and may be caused by a variety of molecular alterations. For example, studies have shown that the formation of breast tumors is closely related to lncRNA [38,39]. Hence, predicting breast cancer-associated lncRNA and identifying lncRNA markers are important for the diagnosis and treatment of breast cancer [39]. In this section, we will implement PMFILDA to discover the potential breast cancer-associated lncRNAs. In addition, as shown in the following Table 3, it is easy to see that 12 of the top 20 breast cancer-related lncRNAs predicted by our model PMFILDA have been confirmed in authoritative databases. For example, MALAT1, HOTAIR and H19 ranked the 1st, 2nd and 3rd in the list of our predicted results respectively, and among them, it is proved that MALAT1 has functional and prognostic significance as a metastasis driver in ER negative and lymph node negative breast cancer [40], HOTAIR will be overexpressed in approximately one quarter of human breast cancers and increased in expression in primary breast tumors and metastases [41], and the down-regulation of H19 will significantly reduce colony formation and anchorage-independent growth of breast and lung cancer cells [42].

Moreover, in recent years, lung cancer is a leading cause of cancer-related deaths worldwide, regardless of gender. According to the disease patterns and treatment strategies, it can be roughly divided into non-small cell lung cancer (NSCLC) and small cell lung cancer [43]. To diagnose and treat lung cancers more effectively, researchers have paid lots of attention to the deregulation of protein-coding genes in the past few decades to identify oncogenes and tumor suppressors [43,44,45]. However, recent studies have shown that lncRNAs play a significant role in the development and progression of lung cancers [43,45]. Hence, in this section, we will implement PMFILDA to infer the potential lung cancer-related lncRNAs. In addition, as illustrated in the following Table 3, it is easy to see that 14 of the top 20 potential lung cancer-related lncRNAs predicted by our model PMFILDA have been confirmed by authoritative biological experiments. For instance, MALAT1, HOTTIP and MEG3 ranked the 3rd, 4th and 5th in the list of our predicted results respectively, and among them, it is identified that MALAT1 is highly correlated with lung cancer metastasis [46,47], will promote lung cancer cell movement by regulating motor-related gene expression [48], and can be an important biomarker for the development of lung cancer metastasis [49]. Additionally, it is demonstrated that through knocking out HOXA13 by RNA interference (siHOXA13), HOTTIP can promote lung cell proliferation, migration, and inhibition of apoptosis, which could serve as a new biomarker and a therapeutic target for NSCLC intervention [50]. Moreover, as for MEG3, it is proved that the down-regulation of MEG3 will enhance cisplatin resistance of lung cancer cells through activation of the WNT/β-catenin signaling pathway [51]. Additionally, the colorectal cancer (CRC) has a high incidence in Western countries in recent years [52], and more and more research indicates that lncRNAs play a significant role in the formation of CRC [44,45]. Hence, in this section, we will implement PMFILDA to predict the potential CRC-related lncRNAs. In addition, as shown in the following Table 2, it is easy to see that eight of the top 20 CRC-related lncRNAs predicted by our model PMFILDA have been confirmed by authoritative biological experiments. For instance, MALAT1, NEAT1 and TUG1 ranked the 2nd, 8th and 10th in the list of our predicted results respectively, and among them, it is identified that MALAT1 may be a potential predictor of tumor metastasis and prognosis, and that the interaction between MALAT1 with SFPQ may be a new therapeutic target for CRC [53]. In addition, it is proved that NEAT1 can be used as an indicator of tumor recurrence and colorectal cancer prognosis [54], and the expression of NEAT1 in CRC may play a carcinogenic role in the differentiation, invasion, and metastasis of CRC, hence, the whole blood NEAT1 expression can be used as a new diagnostic and prognostic biomarker for overall survival in CRC [55]. Moreover, it is demonstrated that the tumor expression of TUG1 plays an important role in colorectal cancer metastasis, and TUG1 can be used as a biomarker or therapeutic target for potential CRC [56].

## 5. Discussion

Increasing research has shown that lncRNAs play a crucial role in the occurrence, formation, diagnosis, treatment, and prognosis of diseases. The discovery of complex disease-associated lncRNAs as biomarkers based on existing biological experiments is not only costly but also requires a large amount of clinical data. Therefore, it is a future trend to integrate potential biological data resources and use developed computers to develop efficient and accurate computational models to predict potential new disease-related lncRNAs. In this paper, we proposed a novel computational model called PMFILDA to predict potential disease-associated lncRNAs. In this model, we first integrated known lncRNA-miRNA associations, miRNA-disease associations, and a small number of known lncRNA-disease associations into a new weighted lncRNA-disease association network. Then, based on the newly constructed association network, through adopting the semantic similarity of the disease, the functional similarity of lncRNA and the KNN algorithm to update the weight network, an lncRNA-disease association matrix $W_{ld}$ can be obtained. Hence, through adopting the probability matrix decomposition scheme to decompose the matrix $W_{ld}$ into the feature matrix *U* of lncRNA and the feature matrix *V* of the disease, we can finally construct our model PMFILDA based on the two feature matrices to predict the potential associations between lncRNAs and diseases. Compared to existing state-of-the-art models, simulation results have demonstrated that our model PMFILDA has better prediction performance. Moreover, case studies of breast cancer, lung cancer and colorectal cancer also indicated that PMFILDA can be used as a superior computational model to predict potential lncRNA-disease associations. However, it is obvious that there are still some biases in our model. When we only use lncRNA-disease associations and regardless of any miRNAs, the performance of PMFILDA may be reduced. To illustrate this situation, we did the following experiment. After processing the data, we obtained 246 pairs of lncRNA-disease associations, including 44 lncRNAs, 68 diseases. Then we performed 100 LOOCVs on the PMFILDA method, and the average AUC value was 0.8111, and the standard deviation was 0.0073. When we used miRNA, the average AUC value was 0.8794 and the standard deviation was 0.0011. The reason for this difference is that when we don${}^{,}$t consider miRNAs, the information we use for lncRNA-disease may be incomplete. There may be some important associations that do not exist in the lncRNA-disease data set. When the miRNA node is added, these important relationships can be re-established. Therefore, in our model, we need to consider not only the lncRNA-disease relationship, but also the nodes that can improve the lncRNA-disease relationship.

## 6. Conclusions

In this study, our major contributions are as follows: Firstly, we constructed a novel weighted lncRNA-disease association network through integrating the known lncRNA-miRNA association network, the known miRNA-disease association network and the known lncRNA-disease association network. Secondly, based on the semantic similarity of disease and the similarity of lncRNA function, we adopted the KNN algorithm to update the newly constructed weighted lncRNA-disease association network. Thirdly, based on the probability matrix decomposition model, we proposed a novel computational model called PMFILDA to predict potential lncRNA-disease associations, which cannot only predict the potential associations between lncRNAs and disease contained in the experimentally validated lncRNA-disease associations, but also predict the potential associations of its elements in unknown datasets. To improve the efficiency of our model, in the future, we plan to integrate more intermediate nodes such as genes to update the weighted lncRNA-disease association network. In addition, we also believe that the results [25,57,58,59,60,61,62,63] of the miRNA-disease association prediction field will promote the development of lncRNA-disease correlation prediction. Moreover, while studying the association prediction of lncRNA-disease, focusing on the research results in other fields will also broaden our horizons.

## Figures and Tables

**Figure 1 genes-10-00126-f001:**
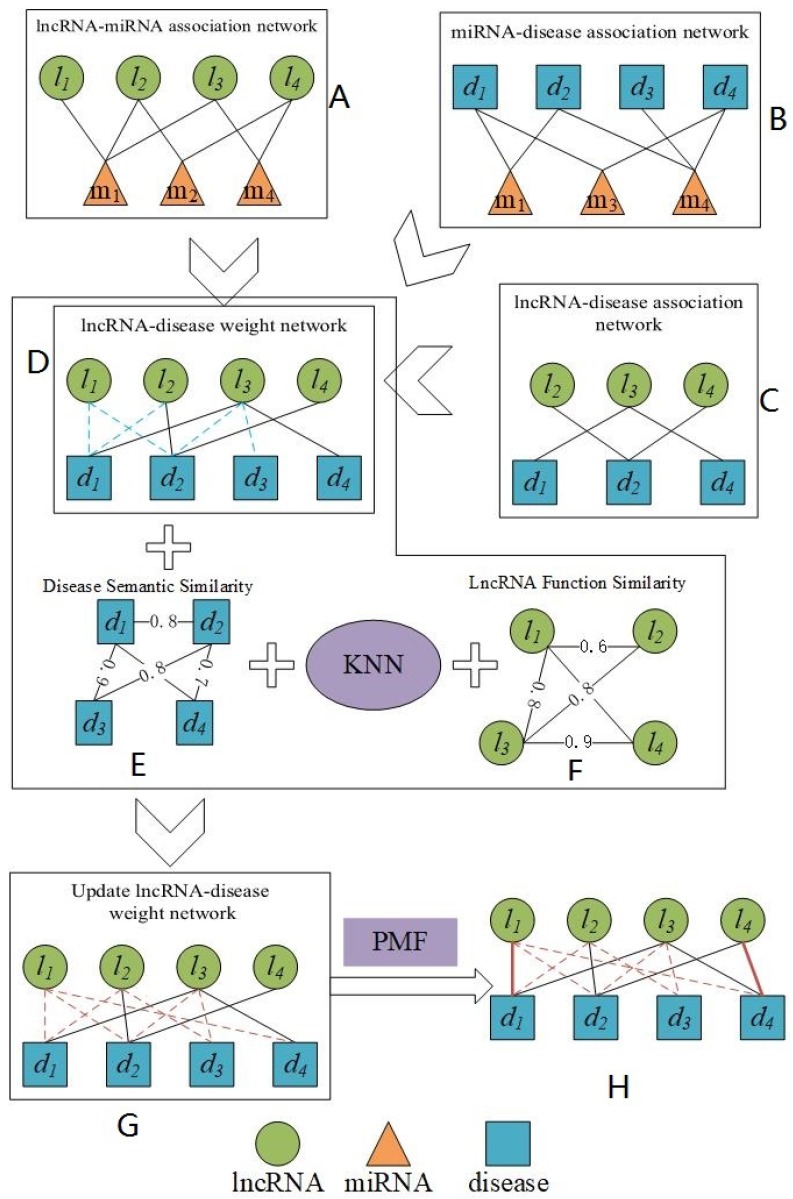
The flowchart of our prediction model of PMFILDA.

**Figure 2 genes-10-00126-f002:**
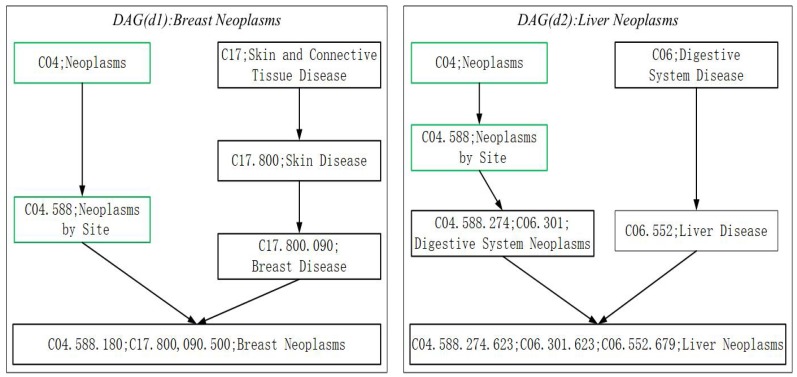
The DAGs of the disease Breast Neoplasms and Liver Neoplasms. In addition, the disease term and its identification numbers are included in corresponding node. The common terms of the two diseases are illustrated by green nodes.

**Figure 3 genes-10-00126-f003:**
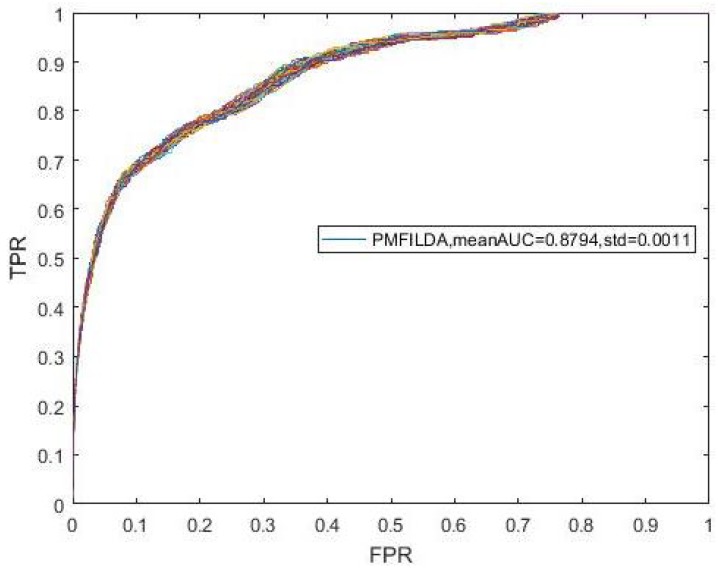
ROC curves for PMFILDA.

**Figure 4 genes-10-00126-f004:**
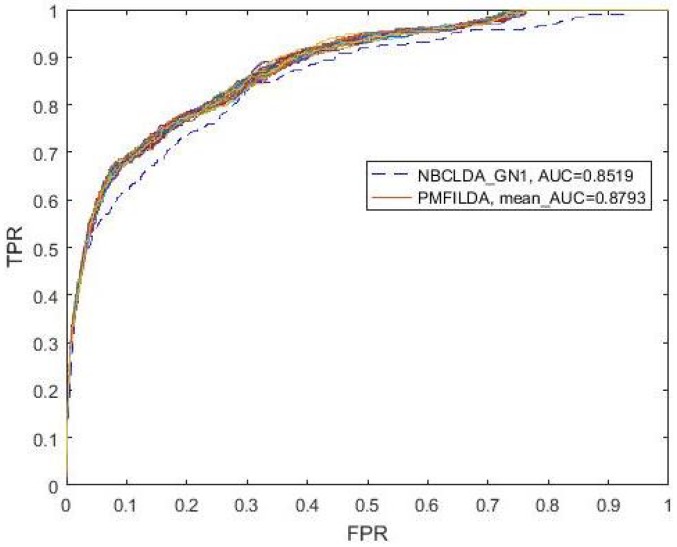
ROC curves and AUC value for $NBCLDA_{GN1}$ and PMFILDA.

**Figure 5 genes-10-00126-f005:**
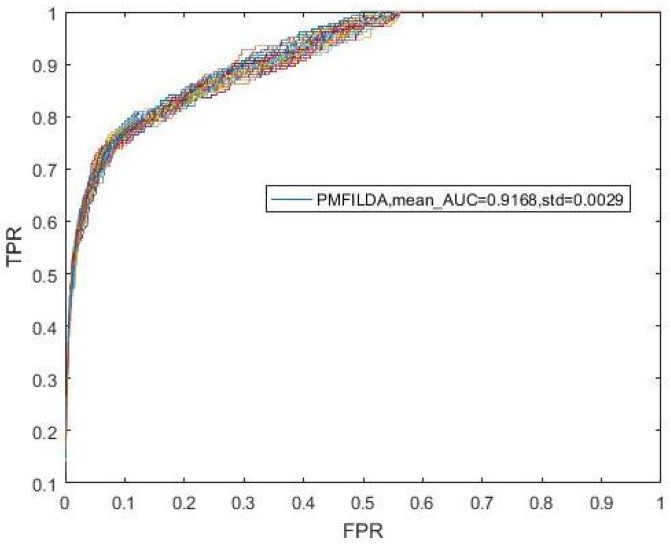
The ROC curve and AUCs of PMFILDA based on the same known 183 lncRNA-disease associations proposed by HGLDA.

**Figure 6 genes-10-00126-f006:**
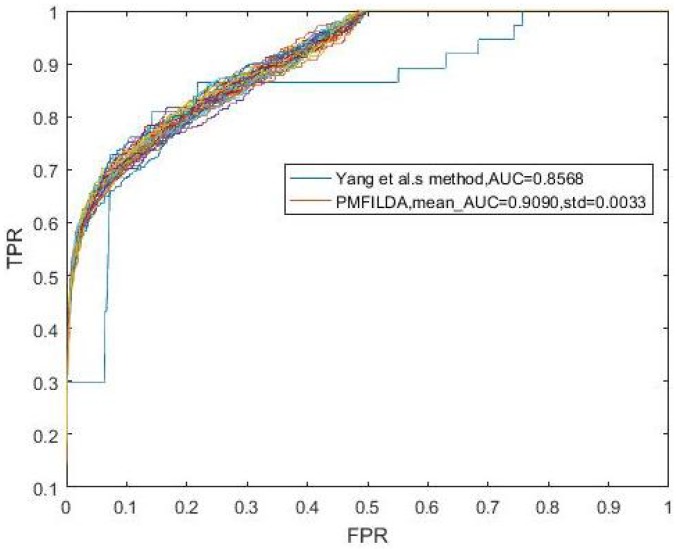
ROC curves and AUCs of PMFILDA and the method proposed by Yang et al.

**Figure 7 genes-10-00126-f007:**
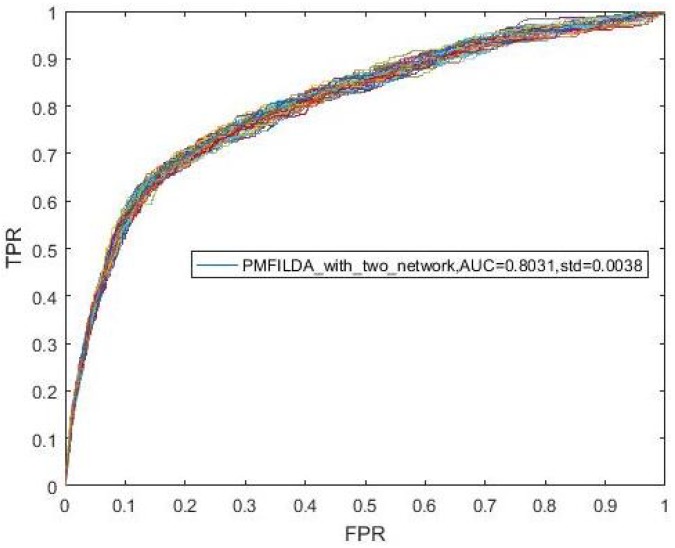
ROC curves and AUCs achieved by PMFILDA based on two association networks.

**Figure 8 genes-10-00126-f008:**
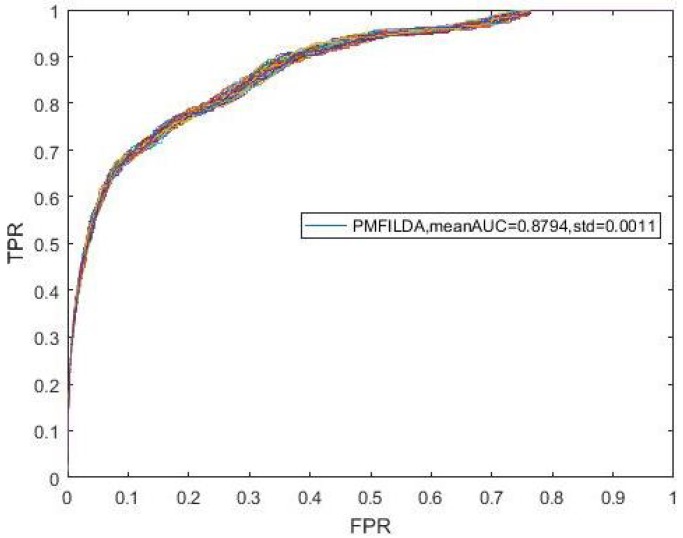
ROC curves and AUCs achieved by PMFILDA based on three association networks.

**Figure 9 genes-10-00126-f009:**
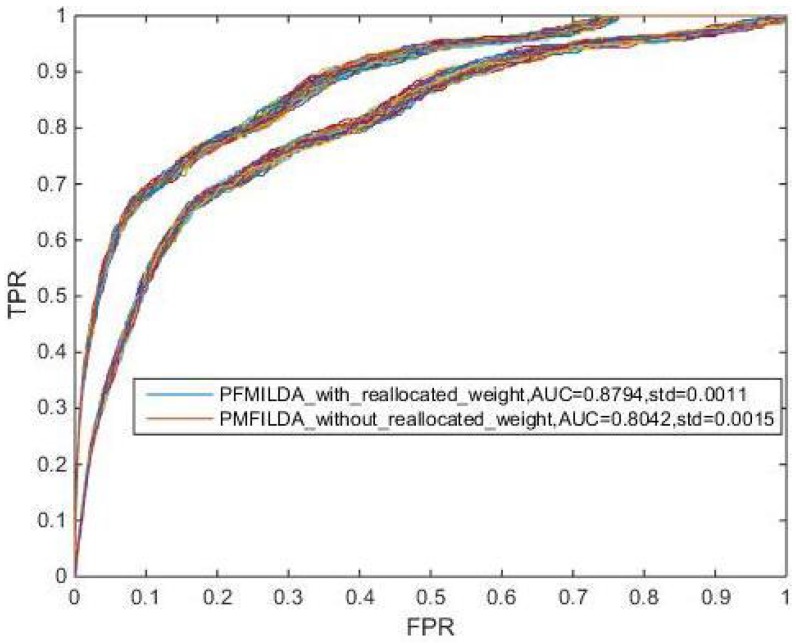
AUCs achieved by PMFILDA in LOOCV while the weight of $G_{LDMN}$ was reallocated or not reallocated respectively.

**Table 1 genes-10-00126-t001:** Comparison of AUCs of PMFILDA with state-of-the-art methods.

Methods	AUCs	Methods	AUCs	Methods	AUCs
PMFILDA	0.8793	PMFILDA	0.9169	PMFILDA	0.9090
$NBCLDA\_GN_{1}$	0.8519	HGLDA	0.8519	Method of Yang et al.	0.8568

**Table 2 genes-10-00126-t002:** Comparison of the effects of KNN and K-means on PMFILDA.

	KNN	K-Means
Mean_AUC	0.8794	0.8589
STD	0.0278	0.0011

**Table 3 genes-10-00126-t003:** The experimentally confirmed lncRNAs in the top 20 potential lncRNAs predicted by PMFILDA in three kinds of case studies.

Diseases	lncRNAs	Evidence (PUBMED)
Breast Cancer	MALAT1	22492512, 22996375, 24499465, 27250026, 27777857, 27191888
Breast Cancer	HOTAIR	24499465, 20930520, 21925379, 20393566, 19182780, 21903344
Breast Cancer	H19	22996375, 21489289, 14729626, 16707459, 21748294, 18794369
Breast Cancer	MEG3	27166155, 14602737, 22393162, 22487937
Breast Cancers	GAS5	27034004, 18836484, 20673990, 22487937, 22664915, 26662314
Breast Cancer	PTPRG-AS1	26409453
Breast Cancer	NEAT1	25417700, 27147820, 21532345, 27556296
Breast Cancer	PVT1	24780616, 17908964, 25122612, 26889781
Breast Cancer	CDKN2B-AS1	17440112, 20956613, 20453838, 20956613
Breast Cancer	TUG1	27791993
Breast Cancer	XIST	17545591, 27248326, 18006640, 19440381, 24141629, 26637364
Breast Cancer	ZFAS1	21460236
Lung Cancer	H19	27186394, 26729200
Lung Cancer	HOTAIR	27186394, 26729200, 24757675, 23668363, 27270317
Lung Cancer	MALAT1	25217850, 20937273, 20937273, 27777857
Lung Cancer	HOTTIP	27347311, 26265284
Lung Cancer	MEG3	14602737, 26059239
Lung Cancer	CDKN2B-AS1	27307748, 26729200, 26453113, 25964559, 25889788
Lung Cancer	GAS5	27631209, 26634743, 24357161
Lung Cancer	CCAT1	25129441
Lung Cancer	XIST	27501756, 26339353
Lung Cancer	CASC2	26790438
Lung Cancer	PVT1	26908628, 26729200, 25400777
Lung Cancer	ZNRD1-AS1	27166266
Lung Cancer	NEAT1	27351135, 27270317, 25889788
Lung Cancer	TUG1	24853421, 27485439
Colorectal Cancer	H19	8564957, 22427002, 11120891, 26989025, 19926638, 26068968
Colorectal Cancer	HOTTIP	26617875, 26678886, 27546609
Colorectal Cancer	XIST	17143621
Colorectal Cancer	NEAT1	26314847, 26552600
Colorectal Cancer	MEG3	25636452, 26934323
Colorectal Cancer	TUG1	26856330, 27421138
Colorectal Cancer	PVT1	26990997, 24196785
Colorectal Cancer	CCAT1	23416875, 26064266, 26823726, 24594601, 23594791, 26752646

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
