# Peer review of "A Probabilistic Matrix Factorization Method for Identifying lncRNA-Disease Associations"

_genes, 2019, doi:10.3390/genes10020126_

Round 1

Reviewer 1 Report

The authors proposed a model based on probability matrix decomposition to predict potential lncRNA-disease correlations. Increasing evidence has shown that lncRNAs play an important role in many complex diseases of human being. In this paper, the authors considered the information of miRNAs, lncRNAs and diseases synthetically, integrated these different data skillfully, and then proposed a probability matrix decomposition based prediction model to explore potential lncRNA-disease relationships. In general, I think that the study is interesting and acceptable, except that a minor revision shall be done. My detailed comments are as follows:

1. In the introduction section, the authors gave a global flow chart, however to some degree the diagram are puzzling. It will be better if the authors can give a description of the flow chart in detail.

2. I noticed that the KNN algorithm was adopted to construct the weighted network, but the author did not describe how the KNN algorithm was adopted in the article. Therefore, I think usage of the KNN algorithm in their prediction model is necessary to be introduced for readers, and it will be better to introduce it in the appendix of this paper.

3. The authors should carefully examine their formulas, for example, in the equations 13 and 14, the equal sign might be "=".

4. Why is the ROC curve of PMRILDA not a curve? I think the authors should give an explanation.

Author Response

Reviewer1:

The authors proposed a model based on probability matrix decomposition to predict potential lncRNA-disease correlations. Increasing evidence has shown that lncRNAs play an important role in many complex diseases of human being. In this paper, the authors considered the information of miRNAs, lncRNAs and diseases synthetically, integrated these different data skillfully, and then proposed a probability matrix decomposition based prediction model to explore potential lncRNA-disease relationships. In general, I think that the study is interesting and acceptable, except that a minor revision shall be done. My detailed comments are as follows:

Comments 1:

In the introduction section, the authors gave a global flow chart, however to some degree the diagram are puzzling. It will be better if the authors can give a description of the flow chart in detail.

Response 1:

We have updated the flowchart and added a detailed description in the paragraph blow the Fig 1. For more details, please see Fig 1 and the paragraph blow it on page 3 of our revised manuscript.

Comments 2:

I noticed that the KNN algorithm was adopted to construct the weighted network, but the author did not describe how the KNN algorithm was adopted in the article. Therefore, I think usage of the KNN algorithm in their prediction model is necessary to be introduced for readers, and it will be better to introduce it in the appendix of this paper.

Response 2:

According to above suggestions, we have added the detailed description of the KNN algorithm in the appendix A. For more details, please see the description in the appendix A of our revised manuscript.

Comments 3:

The authors should carefully examine their formulas, for example, in the equations 13 and 14, the equal sign might be "=".

Response 3:

We have checked all formulas carefully and corrected all errors including the errors mentioned above.

Comments 4:

Why is the ROC curve of PMRILDA not a curve? I think the authors should give an explanation.

Response 4:

During simulation, in order to eliminate the random errors caused by the random initialization of the matrices U and V, we repeated our experiments for 100 times, therefore, there are many ROC curves instead of a ROC curve

Reviewer 2 Report

LncRNA is a new class of non-coding RNA molecules that are thought to be important for the development of human diseases. Considering high costs of traditional bio-experiments, it is highly promising to develop computational models to accelerate the study of associations between lncRNAs and diseases. The manuscript described a novel approach called PMFILDA to model the associations between lncRNAs and diseases through using probability matrix decomposition. The method is technically fine and can be very useful to the community. I have only the following minor comments: 

1. The authors re-allocated the weight of G-LDMN. What kinds of effects would it have if it was not reallocated? In addition, there are lots of classification algorithms, why did the authors preferred the relatively simple KNN algorithm instead of other more complicated classification algorithms?

2. Some latest studies in this research direction are missing and should be discussed and cited in the manuscript.

3. The manuscript needs to be substantially rewritten. There are typos and grammatical errors throughout the text, e.g. in the part of abstract, the diseases adopted by the authors are given as “breast cancer, lung cancer and colorectal cancer”, however, in Table 2, the diseases changed to be "Breast Neoplasms, Lung Neoplasms and Ovarian Neoplasms". I therefore strongly suggest the authors carefully go through their manuscript and fix all these errors.

Author Response

Dear Editors and Reviewers,

First of all, thanks very much for your great help. According to the valuable comments, we modified our paper carefully. Our point-to-point responses are given as follows:

Reviewer 2:

LncRNA is a new class of non-coding RNA molecules that are thought to be important for the development of human diseases. Considering high costs of traditional bio-experiments, it is highly promising to develop computational models to accelerate the study of associations between lncRNAs and diseases. The manuscript described a novel approach called PMFILDA to model the associations between lncRNAs and diseases through using probability matrix decomposition. The method is technically fine and can be very useful to the community. I have only the following minor comments:

Comments 1:

The authors re-allocated the weight of G-LDMN. What kinds of effects would it have if it was not reallocated? In addition, there are lots of classification algorithms, why did the authors preferred the relatively simple KNN algorithm instead of other more complicated classification algorithms?

Response 1:

According to above comments, we implemented PMFILDA in LOOCV while the weight of G-LDMN was reallocated or not reallocated respectively, and the simulation results were shown in the following Fig 9. From observing the Fig 9, it is easy to see that PMFILDA could achieve an average AUC of 0.8794 while the weight of G-LDMN was reallocated, however, while the weight of G-LDMN was not reallocated, PMFILDA can achieve an average AUC of 0.8042 only, which demonstrated that it can improve the performance of our model through adopting the KNN algorithm to re-allocate the weight of G-LDMN. For more details, please see the section 4.3 on page 13 of our revised manuscript.

Fig 9. AUCs achieved by PMFILDA in LOOCV while the weight of G-LDMN was reallocated or not reallocated respectively

And in addition, the reason to choose KNN is that KNN is not only very simple, but also is a non-parametric method, and moreover, its generalization error rate is less than twice the error rate of Bayesian optimal classifier. Moreover, in order to estimate the impacts of other algorithms, we selected the K-means algorithm for further testing. After performing LOOCV 100 times, we presented the simulation results in the following table 1, and from observing the results in table 2, it is easy to see that KNN algorithm has better performance than K-means algorithm. For more details, please see the section 4.3 on page 13 of our revised manuscript.

Table 2: Comparison of the effects of KNN and K-means on PMFILDA

KNN

K-means

Mean_AUC

0.8794

0.8589

STD

0.0278

0.0011

Comments 2:

Some latest studies in this research direction are missing and should be discussed and cited in the manuscript.

Response 2:

According to above comments, we have added some of the latest studies in related research directions and at the same time, we have discussed and cited these latest studies in our revised manuscript as well. For example, (1) as for the research direction of lncRNA function prediction, we have added references 11 and 12, and discussed and cited them on the first page of our revised manuscript. (2) As for the research direction of the application of Matrix Factorization in the field of bioinformatics, we have added references 23, 24, 25, and discussed and cited them on page 2 of our revised manuscript. (3) As for the research direction of the application of KNN algorithm in bioinformatics, we have added a reference 26, and discussed and cited it on page 3 of our revised manuscript. In addition, Considering that known miRNA-disease correlation was adopted in our novel model, therefore, as for the research direction of miRNA-disease association prediction, we further explored the contribution of miRNAs and the importance of miRNAs in our prediction model (For more details, please see section 5 on page 16 of our revised manuscript). And moreover, in section 6, we have added a reference 24, 56, 57 and 58, and discussed and cited them on page 17 of our revised manuscript.

Comments 3:

The manuscript needs to be substantially rewritten. There are typos and grammatical errors throughout the text, e.g. in the part of abstract, the diseases adopted by the authors are given as “breast cancer, lung cancer and colorectal cancer”, however, in Table 2, the diseases changed to be "Breast Neoplasms, Lung Neoplasms and Ovarian Neoplasms". I therefore strongly suggest the authors carefully go through their manuscript and fix all these errors.

Response 3:

According to above comments, we have checked our manuscript carefully and revised all typos and grammatical errors in our revised manuscript.

Reviewer 3 Report

In this study, Xuan et al., developed a novel computational model called PMFILDA to predict potential lncRNA-disease associations. They constructed three association networks 1) lncRNA-miRNA association networks, 2) miRNA-disease association networks, and 3) lncRNA-disease association networks, and integrated these to improve the accuracy of lncRNA-disease association predictions. LncRNAs are a novel class of biomolecules, and the least understood the type of RNA molecules. As of May 2018, the genomic database Ensembl holds about 15,000 human lncRNA sequences. However, functional information is available only for a few lncRNAs. There is now a vast demand for computational methods to accelerate lncRNA research. Overall the paper is well written but requires a major revision before publication in genes.

Major Comments

1)      It will be useful for the readers to show the contribution from individual association network. For example, one could use only two out three association networks to construct lncRNA-disease weight networks and thereby test the contribution from each parameter.

2)      The choice of KNN has not been mentioned clearly. I wonder if the authors examined other algorithms. Testing other algorithms might improve the robustness.

3)      I am also curious about how PMFILDA performs on lncRNAs-disease associations that do not involve miRNAs. Are there any examples or outliers where PMFILDA performed poorly because there are no miRNAs involved in that lncRNA-disease association? This would be helpful to add in the discussion.

Author Response

Reviewer 3:

In this study, Xuan et al., developed a novel computational model called PMFILDA to predict potential lncRNA-disease associations. They constructed three association networks 1) lncRNA-miRNA association networks, 2) miRNA-disease association networks, and 3) lncRNA-disease association networks, and integrated these to improve the accuracy of lncRNA-disease association predictions. LncRNAs are a novel class of biomolecules, and the least understood the type of RNA molecules. As of May 2018, the genomic database Ensembl holds about 15,000 human lncRNA sequences. However, functional information is available only for a few lncRNAs. There is now a vast demand for computational methods to accelerate lncRNA research. Overall the paper is well written but requires a major revision before publication in genes.

Comments 1:

It will be useful for the readers to show the contribution from individual association network. For example, one could use only two out three association networks to construct lncRNA-disease weight networks and thereby test the contribution from each parameter.

Response 1:

According to above comments, we only used the lncRNA-miRNA and miRNA-disease associations to construct the lncRNA-disease weight network. And then, based on the steps in section 3.2, we finally obtained 304 lncRNA-disease associations including 60 lncRNAs and 73 diseases. Thereafter, we further obtained the corresponding weight matrix W_ld, and then performed LOOCV 100 times on the PMFILDA. The results were shown in the following Figs, and from observing these two Figs, obviously, the AUC value achieved by PMFILDA based on three association networks can be increased by 0.0763 than the AUC value achieved by PMFILDA based on two association networks. For more details, please see the Fig 7 and Fig 8 in our revised manuscript.

Fig 7. ROC curves and AUCs achieved by PMFILDA based on two association networks

Fig 8. ROC curves and AUCs achieved by PMFILDA based on three association networks.

Comments 2:

The choice of KNN has not been mentioned clearly. I wonder if the authors examined other algorithms. Testing other algorithms might improve the robustness.

Response 2:

The reasons to choose KNN are as follows: considering that known lncRNA-disease associations are very sparse, which may cause that there exist some lncRNAs having no associations with any diseases, or some diseases having no associations with any lncRNAs. Hence, some potential associations between predicted lncRNAs and diseases will be invalid. Therefore, in this paper, we will rebuild the weight matrix W_ld based on KNN algorithm to solve this kind of problem. For more details, please see section 3.3.3 on page 6 and section 4.3 on page 13 of our revised manuscript.

And in addition, in order to estimate the impacts of other algorithms, we selected the K-means algorithm for further testing. After performing LOOCV 100 times, we presented the simulation results in the following table 2, and from observing the results in table 2, it is easy to see that it is easy to see that the performance of KNN is better than K-means.

Table 2: Comparison of the effects of KNN and K-means on PMFILDA

KNN

K-means

Mean_AUC

0.8794

0.8589

STD

0.0278

0.0011

Comments 3:

I am also curious about how PMFILDA performs on lncRNAs-disease associations that do not involve miRNAs. Are there any examples or outliers where PMFILDA performed poorly because there are no miRNAs involved in that lncRNA-disease association? This would be helpful to add in the discussion.

Response 3:

According to above comments, in order to evaluate the prediction performance of PMFILDA based on lncRNAs-disease associations that do not involve miRNAs, we only used lncRNA-disease associations to construct the lncRNA-disease weight network. And after preprocessing of the data, we finally obtained 246 pairs of lncRNA-miRNA associations including 44 lncRNAs and 68 diseases. Then we performed LOOCV 100 times on the PMFILDA method, and as a result, the average AUC value achieved by PMFILDA was 0.8111, and corrsponding standard deviation was 0.0073. While miRNAs are included in the construction of the lncRNA-disease weight network, the average AUC value achieved by PMFILDA would be 0.8794, and corrsponding standard deviation would be 0.0011. Through analysis, it is easy to know that while miRNAs are not considered, there may be some important associations that do not exist in the lncRNA-disease network, however, after miRNAs are added into the network, then those important relationships can be re-established. For more details, please see discussion on page 17 of our revised manuscript.

Round 2

Reviewer 1 Report

Reference fromat should be checked carefully. There are many small mistakes. because for data, lncRNA and miRNA are methodically related, the difference is the size of the data set.-----This sentence is not toally correct.

Author Response

Point 1:

Reference format should be checked carefully. There are many small mistakes. because for data, lncRNA and miRNA are methodically related, the difference is the size of the data set.-----This sentence is not toally correct.

Response 1:

We have carefully revised the format of references, including the last and first names of authors, the volume number, year of publication and page numbers of journals (For more details, please see the section of References from page 17 to page 20 of our revised manuscript). Moreover, we have deleted the sentence "because for data, lncRNA and miRNA are methodically related, the difference is the size of the data set" (For more details, please see section 6 on page 17 of our revised manuscript). Besides, we have further added some of the latest literatures on prediction of potential lncRNA-disease associations and miRNA-disease associations, such as references 13 and 60-63 on pages 18 and 20 in our revised manuscript.
